# Lessons Learned from Natural Disasters around Digital Health Technologies and Delivering Quality Healthcare

**DOI:** 10.3390/ijerph20054542

**Published:** 2023-03-03

**Authors:** Zerina Lokmic-Tomkins, Dinesh Bhandari, Chris Bain, Ann Borda, Timothy Charles Kariotis, David Reser

**Affiliations:** 1School of Nursing and Midwifery, Monash University, 35 Rainforest Walk, Clayton, Melbourne, VIC 3800, Australia; 2Digital Health Theme, Department of Human-Centered Computing, Faculty of Information Technology, Monash University, Melbourne, VIC 3800, Australia; 3Melbourne Medical School, The University of Melbourne, Parkville, VIC 3010, Australia; 4Department of Information Studies, University College London, London WC1E 6BT, UK; 5School of Computing and Information System, The University of Melbourne, Melbourne, VIC 3010, Australia; 6Melbourne School of Government, The University of Melbourne, Melbourne, VIC 3010, Australia; 7Graduate Entry Medicine Program, Monash Rural Health-Churchill, Churchill, VIC 3842, Australia

**Keywords:** climate change, natural disaster, digital health technology, healthcare, citizen science, resilience

## Abstract

As climate change drives increased intensity, duration and severity of weather-related events that can lead to natural disasters and mass casualties, innovative approaches are needed to develop climate-resilient healthcare systems that can deliver safe, quality healthcare under non-optimal conditions, especially in remote or underserved areas. Digital health technologies are touted as a potential contributor to healthcare climate change adaptation and mitigation, through improved access to healthcare, reduced inefficiencies, reduced costs, and increased portability of patient information. Under normal operating conditions, these systems are employed to deliver personalised healthcare and better patient and consumer involvement in their health and well-being. During the COVID-19 pandemic, digital health technologies were rapidly implemented on a mass scale in many settings to deliver healthcare in compliance with public health interventions, including lockdowns. However, the resilience and effectiveness of digital health technologies in the face of the increasing frequency and severity of natural disasters remain to be determined. In this review, using the mixed-methods review methodology, we seek to map what is known about digital health resilience in the context of natural disasters using case studies to demonstrate what works and what does not and to propose future directions to build climate-resilient digital health interventions.

## 1. Introduction

The sudden emergence of the COVID-19 pandemic and subsequent public health measures enacted to minimise COVID-19 transmission were accompanied by the rapid, worldwide deployment of digital health technologies (DHT) across all healthcare settings. This deployment facilitated safer patient screening and management; minimised exposure of healthcare workers and the public to the COVID-19 virus; improved modelling of disease spread; and supported communication between patients and their families or healthcare workers [1]. Telehealth became a primary communication conduit for locked-down or isolated patients, critically ill individuals, and healthcare professionals [2]. Other DHTs, including artificial intelligence, mobile health apps, big data analytic technology, 5G internet, and the Internet of Things (IoT), have increased in availability in recent years. The use of these technologies will be essential to healthcare delivery going forward, especially under adverse conditions. Finally, consumers have appropriated many digital technologies to support their response to the COVID-19 pandemic, for example, by using social media to find health information or emotional support [3].

In this review, we define what we mean by DHTs, and examine the advantages and liabilities DHTs confer in the context of likely climate-related disasters. We aim to explore the prior planning requirements, infrastructural requirements, and vulnerabilities of DHTs as means of delivering effective care to the affected populations [4]. Other digital technologies that may impact and play a key role in the delivery of care during natural disasters but are not considered DHTs, such as interactive maps, databases, text-bots, and drones, will briefly be discussed.

Given the rapid development cycle and adoption of DHTs, and the absence of formal frameworks around how DHTs are used in natural disaster situations, it is difficult to accurately predict how DHTs will affect the delivery of quality healthcare during natural disasters. This uncertainty is highlighted in disasters precipitated by extreme weather events, especially during and in the immediate aftermath of catastrophic events, where management of acute health emergencies is paramount. Moreover, technology not designed for healthcare might be appropriated as a DHT during a disaster response, if formal technology is not available or appropriate [5], and the resilience of digital health technology (DHT) systems to repeated or knock-on disasters (e.g., widespread flooding or landslides following a massive fire event) impacting the same healthcare facility or delivery catchment still needs to be better understood. There remains a digital divide that plagues society and hinders access to health services, which will also shape the use of DHTs in disaster situations [6]. Further, we know little about community-led digital health responses during a disaster and how digital technology, such as social media, might be appropriated to support people’s health during and after a disaster. Notwithstanding, DHTs have played a critical and expanding role in response to natural disasters worldwide, including those of human, climatic, biological, and geophysical origin.

The lessons learned and vulnerabilities exposed by disaster events such as Nepal’s 2015 earthquake and the 2019–2020 Australian bushfires will shape healthcare delivery in the future, especially in rural, remote, and underserved communities. Therefore, we must examine how DHTs can be relied on in the delivery of quality healthcare interventions during natural disasters and assist in preparation and recovery efforts associated with these events. This issue is discussed with respect to lessons learned from past natural disasters about how to deliver safe, high-quality care, and mitigation of associated risks, to build a resilient digital health system and digital healthcare models. We commence this mixed-methods review (see Section 5.1) by defining digital health, digital health technologies, and digital health interventions before using case studies to identify areas for further consideration.

## 2. Digital Health

Digital health is defined as ‘an umbrella term referring to a range of technologies that can be used to treat patients and collect and share a person’s health information’ [7]. A non-comprehensive list of DHTs includes tools used to deliver digital health interventions, defined as health services delivered electronically, either formally or informally [8], including communications; information storage and access; predictive and analytical systems; remote sensing; and mobile systems. DHTs can include tools used by clinicians and health services to deliver and manage care, and tools used by health service users and patients/consumers to manage their own health and wellness. Delivery of digital health interventions depends on the type of digital health technologies used, infrastructure, security, and the accessibility of the technologies. It is also important to note that DHTs are used to support broader needs of healthcare such as making appointments, which are as equally important as digital health interventions.

DHT systems broadly implemented for healthcare delivery include telehealth, electronic medical records (EMR), electronic prescribing, electronic referrals, and mobile health (mHealth); and applications such as Short Message Service (SMS), wellness apps, wearables (trackers and monitors), and sensors [9]. Each of these encompasses a variety of settings and applications. For instance, telehealth services can range from simple phone calls to complex video systems connecting multiple stakeholders [10]. The use of these technologies may be similar or vastly different across disaster and non-disaster contexts. There may also be new uses for these technologies, and the need for different technologies in disaster scenarios. Currently, there is no clear framework or evidence-based guidelines for how these DHTs can and should be used in disaster situations.

Critically, digital health interventions are seen as a key component of the mitigation strategy to address healthcare’s 5% contribution to global greenhouse gas emissions [11]. For example, DHTs can facilitate the integration of climate and health data for a dashboard visualisation to monitor impacts, modelling to predict climate-related impacts on health, and as decision support tools to provide alerts for potential heat events [12]. However, to meet this goal, DHTs needs to be designed to be environmentally sustainable [13,14], as they have a carbon footprint and generate a large volume of environmentally damaging e-waste, which is often sub-optimally managed [15,16]. Though it is not the purpose of this paper, it is critical to recognise that though DHTs can contribute to disaster responses, DHTs need to be designed in a way that does not further contribute to the climate disaster [14].

As alluded to above, many terms associated with DHTs are used interchangeably, causing confusion around the definition, implementation, and regulation of digital health systems of care. Thus, we define below what we mean by each of the DHTs in the context of this review. It is important to note that these DHTs do not function in isolation; rather, they are often connected to form a digital ecosystem purposed to deliver safe and quality healthcare and support daily healthcare operations.

### 2.1. Telehealth and Telemedicine

Telehealth is defined as *‘the delivery and facilitation of health and health-related services, including medical care, provider and patient education, health information services, and self-care via telecommunications and digital communication technologies’* [17]. Other terms used for telehealth are telemedicine and virtual care, although the literature indicates that these are not synonymous. Telehealth includes all components and activities of healthcare and healthcare systems conducted via telecommunication technology, while telemedicine refers only to the practice of medicine via remote means [17]. Virtual care is broader than telehealth and telemedicine, including smartphone apps, and mobile and internet-connected devices for reporting, collection, transmission, and assessment of patient data [18]. Delaigue et al. examined the Medicine Sans Frontiers telemedicine service from July 2010 to June 2017 [19]. The service allows Medicine Sans Frontiers field officers access to a network of specialists at Medicine Sans Frontiers headquarters who cover the areas of paediatrics, surgery, infectious diseases, internal medicine, nutrition, anaesthesia, and obstetrics. The study showed that offering direct specialist expertise in low-resource settings improved the management of patients and provided additional value to physicians in the field through educational opportunities [19]. Other applications of telehealth, telemedicine, and virtual care are described in Table 1 and Table 2.

### 2.2. Electronic Health Records

Terms such as electronic medical records (EMR), electronic health records (EHR), electronic patient records (EPR) and personal health records (PHRs) are also often used interchangeably. For our purposes, an EMR refers to ‘*an electronic record of health-related information on an individual that can be created, gathered, managed, and consulted by authorised clinicians and staff within one health care organisation*’ [20]. In addition to being confined to a single organisation, EMRs often lack nationally recognised interoperability standards, which are a key feature of EHRs. The National Alliance for Health Information Technology defines an EHR as ‘*An electronic record of health-related information on an individual that conforms to nationally recognized interoperability standards and that can be created, managed, and consulted by authorised clinicians and staff across more than one health care organisation*’ [20]. An example of this would be My Health Records, Australia’s national EHR system [21]. My Health Record allows treating clinicians to upload a summary of a patient’s health information for future treating clinicians to access. A PHR is defined as ‘*an electronic record of health-related information on an individual that conforms to nationally recognized interoperability standards and that can be drawn from multiple sources while being managed, shared, and controlled by the individual.*’ [20]. These technologies may sit alongside one another, where a healthcare institution might have an EMR that feeds into both an EHR and PHR.

The overall value of these records is the accessibility of information to multiple users as defined by the context of that technology (for example, a single institution). In a disaster setting, the availability and portability of these records are advantages with respect to continuity of care and management of ongoing health conditions for displaced populations. One example of the value of EHRs was their use to support the continuity of care for evacuated veterans after Hurricane Katrina in the United States [22].

### 2.3. Electronic Prescribing

Electronic prescribing permits the legal prescribing, dispensing, and claiming of medicines in electronic format without requiring a paper prescription [23]. Electronic prescribing became more prevalent as the COVID-19 pandemic necessitated changes of practice in the interest of protecting public health [23,24].

### 2.4. Electronic Referrals

Electronic referrals, or e-referrals, are systems designed to automate the paper-based referral process, in which appointments and other information pertinent to the requested appointment are transferred between two or more healthcare providers [25]. The value of this system is that it is faster than paper-based referrals, with some evidence suggesting it improves access to specialist care, reduces waiting times, and improves communication between primary care providers and specialists through improved quality of the referral process [26].

### 2.5. mHealth

Mobile health, abbreviated as m-Health or mHealth, refers to the practice of medicine and public health supported by mobile devices, such as smartphones, tablet computers, personal digital assistants, and wearable devices, e.g., smartwatches, that can provide cost-effective health services and facilitate data collection for sharing between the healthcare provider and the consumer [27]. The value of evidence-based *mHealth* is that it is convenient, provides more accessible communication between the patient and care provider, provides secure messaging, and can be designed for patient data to be uploaded to an EMR directly, thus ensuring continuity of care [28]. The most successful example of mHealth is text-message-based health interventions via SMS that provide reminders at any time, place, or setting via phone [29].

Examples of mHealth applications include remote patient monitoring to collect patient data and transmit that data to healthcare settings, where healthcare professionals can review it; patient education on health promotion or preventative healthcare; disease surveillance; treatment support; and chronic disease management. However, many mHealth apps, including wellness apps, have less than ideal privacy policies, and may need better security systems and notification systems in case of a data breach [28].

There is also a burgeoning consumer market for mHealth applications which the health system has not designed. These can include applications ranging from step counters and sleep-tracking applications to mindfulness tools. Increasingly, there is a blurring of the lines between consumers’ mHealth applications and those designed and delivered by the healthcare system. This situation raises questions about how health services utilise data, information, and findings from mHealth applications that they have not prescribed or that may have been designed by a technology company [30]. This is particularly important in disasters where people with no established health record and who might be accessing health services for the first time may only have the data on their phone to give health services a picture of their health and wellness.

### 2.6. Artificial (Augmented) Intelligence and Machine Learning

While many definitions of artificial or augmented intelligence (AI) exist, for this review, AI is defined as an interdisciplinary field of science concerned with developing *‘methods of achieving goals in situations in which the information available has a certain complex character. The methods that have to be used are related to the problem presented by the situation and are similar whether the problem solver is human, a Martian, or a computer program*’ [31]. Machine learning is an application of AI that allows quick searches of disparate databases and other sources, facilitating the integration of information that might be viewed as unrelated or irrelevant. In the context of climate change and natural disasters, this process has applications for predicting areas of impact through terrain and infrastructure mapping, monitoring and managing of large, displaced populations, and logistical management of care delivery. At a more general level, AI is used for improved training and simulation for healthcare workers, computer-aided diagnosis and consultation, and decision analysis. These processes, in turn, can improve care delivery in areas experiencing staff shortages, inconvenient hours, and financial pressures [32].

Though AI has great promise for improving healthcare delivery and management, there are actual and perceived risks and vulnerabilities that must be considered as these systems are developed and adapted for use in healthcare under normal and non-optimal operating conditions (see Table 1) [33]. A recent scoping review by Gunasekaran and colleagues (2021), which examined applications of AI and telehealth during the COVID-19 pandemic, indicated that these issues are understudied with respect to AI compared to DHTs such as telehealth and mobile apps, which may slow adoption of AI relative to other DHTs [34].

**Table 1 ijerph-20-04542-t001:** Robots and their application in delivery of healthcare.

Type of Robotics	Definition	Examples of Applications in Healthcare Industry
**Telerobots**	Semi-autonomous robots that can be controlled from a distance by human operators using a wireless network, televisions, or tethered connections.	A popular telerobot platform (da Vinci system) has been in use for urological and cardiac surgery since the beginning of the 21st century [35]. Telerobots have great potential for use in locating survivors and human rescue in the aftermath of disasters, as well as delivering short-distance medical services to disaster victims [36].
**Collaborative robots**	Service robots that can co-exist in close proximity with humans, maintaining/ensuring a high level of human safety during operation.	Collaborative robots are in wide use in healthcare settings for laboratory testing of biological samples [37]. Collaborative multi-robot systems have potential for use in search and rescue operations during disasters [38].
**Autonomous robots**	Robotic systems capable of independent actions with minimal or no interaction from human operators.	Autonomous robots have been used for various surgical procedures since the mid-1980s [39]. Autonomous robots like flying drones can be used in disaster settings for search or rescue operations, and for delivery of essential medical supplies (e.g., medicines or sterile equipment).
**Social robots**	Artificial intelligence systems that are capable of interaction and communication with humans and their surrounding environment.	Social robots (e.g., PETRA [40]), capable of detecting signs of diabetes and hypothyroidism, have been developed by pharmaceutical companies to support pre-screening for these diseases. In disaster settings, social robots can be used to provide mental health support to victims undergoing rehabilitation [41].
**Wearable robots**	Human-worn smart electronic devices that provide information about body signals, such as vital signs or physical activity, to support or reinforce capabilities of the users.	Surface electromyography is in use for limb assessments, rehabilitation, and assistance [42]. Wearable robots can be used for triaging disaster causalities in low-resource settings [43].

### 2.7. Internet of Things

The Internet of Things (IoT) describes the network of physical objects or devices embedded with sensors, software, and other technologies to connect and exchange data with other devices and systems through the internet to deliver healthcare [44]. The applications of the IoT include wearables and home monitoring equipment, which provide person-centred telemetry that enables the users and their carers to review data and adjust their daily healthcare needs. These data can then be uploaded to hospital networks for healthcare providers to review and provide timely treatment advice. Alert mechanisms can be programmed to alert the user, their carer, and healthcare providers if immediate actions are required. The accuracy and usability of these systems are continually improving, as is the range of body processes that can be monitored [45]. In addition to vital signs such as heart rate, blood pressure, and temperature, devices for monitoring metabolic and disease processes, such as blood glucose [46], seizure activity [47], and cardiac rhythms, are either in use or in the late stages of development [48].

The advantage of this approach in disaster situations is that continuity of information can be maintained even if patients are evacuated or otherwise displaced, without the need for transfer of paper records or new diagnostic evaluation. Other examples of IoT-driven healthcare include real-time medical equipment tracking, infection prevention measures, and asset management, such as inventory control, environmental monitoring (temperature and humidity), and logistics and supply chain management [49,50]. The value of the IoT rests in the potential of seamless communication and data collection that would permit using IoT data to better manage healthcare-related processes and operations, including optimisation of productivity and efficiency, the discovery of new models of healthcare that would lead to value-based care, and reduction of wastage [51]. In the climate-related disaster scenario, consideration must be given to the ruggedness of recording and monitoring equipment (e.g., battery life, water resistance, and operating temperature range) and the availability of communications infrastructure for data transfer.

### 2.8. Robotics

Robot technology featured heavily during the COVID-19 pandemic, mainly in the context of infection prevention—for example, cleaning floors—and performing repetitive tasks such as delivering food, sanitation, and information sharing to free up healthcare workers to focus on patient care [52]. These self-contained mechanical DHTs, which operate autonomously in task-oriented ways and alongside people, can be classified as telerobots; collaborative robots; autonomous robots; social robots; and wearable robots. Potential and actual applications for these robots are described in Table 2. The Internet of Robotic Things (IoRT) integrates robots with sensors and IoT devices to provide real-time health information, thus reducing the risk of human errors [53].

Robotic systems play a role in delivering healthcare under normal and post-disaster conditions, and these applications overlap and interact with the AI and IoT processes described above. The development of advanced surgical robots which can be remotely operated using telehealth systems has promise for surgical intervention in remote and underserved areas. Although the technology in this area has sometimes failed to keep up with hype and speculation [54], in future cases where healthcare personnel are unable to reach disaster-stricken areas, this technology may allow for the delivery of otherwise unavailable care. Extensive research and development will be required for this to be genuinely feasible [55]. More prosaically, drone technology has been employed for surveillance, rescue planning, and disaster management following recent floods in Germany [56].

Uncrewed vehicle systems are also in development for various disaster management scenarios, including search and rescue and supply delivery to inaccessible areas [57]. Accessing difficult and dangerous terrain via these uncrewed vehicles could serve healthcare delivery in those areas [58]. Moreover, planning for healthcare delivery under adverse conditions should include these technologies for training and simulations to improve the coordination of healthcare delivery in a climate disaster-prone future.

### 2.9. Wearables (Trackers and Monitors)

Wearable technology, also known as wearables, encompasses any electronic device connected to the internet and designed to be worn on the user’s body, offering means to capture, send, and receive data specific to the user’s needs [59]. Wearables are also an essential category of the IoT [60]. Examples of wearable technology include smartwatches, wristbands, smart rings, smart glasses, smart patches, smart lenses, smart textiles, jewellery, face masks, and electronic epidermal tattoos [59]. One example of wearable technology adapted for healthcare use is a non-invasive smart patch designed to detect early signs of breast cancer and transmit the information to healthcare providers for assessment [61]. Another example is integration of smartwatch data to capture heart rate, sleep time, and steps taken in a day, which are then fed to a trained algorithm for detection of early stages of infection [59]. Similar to mHealth applications, wearables can arise both from the healthcare system and the consumer market. Though the line between these types of wearables is blurred, the healthcare system still faces barriers to adoption of consumer health technology [62]. In the context of healthcare delivery in natural disasters, this technology may assist search and rescue by remote monitoring and finding of survivors, or by providing tailored early alerts to users in certain areas [63,64].

### 2.10. Digital Markers and Sensors

Digital biomarkers refer specifically to patient-generated physiological and behavioural measures captured by connected DHTs that can be used to diagnose, explain, treat, or monitor a health condition or predict health outcomes. Networks of wireless sensors and transmitters can be deployed for terrain monitoring, air and water quality testing, and search and rescue applications in conjunction with robotic systems described in the previous section. These sensor networks consist of low-cost wireless, solar, or battery-powered detection equipment linked to central or field-deployable monitoring stations [51].

Detection targets include, but are not limited to, position, vibration, temperature, moisture, particulates, and biological or chemical contaminants. While many are still in the research or development stages, the rollout of these systems will likely proceed concurrently with the predicted increase in climate-related threats to individual and public health. The applications for sensor networks that are useful for healthcare workers in disaster settings include monitoring air quality (e.g., post-fire or building collapse), monitoring water levels and portability, and monitoring the integrity of packaged medical supplies [49]. The supply chain for vital medical supplies is an identified area of vulnerability during times of disaster and pandemics [50]. Accordingly, training and management of healthcare personnel in vulnerable areas should include consideration of the refinement and applications of such systems.

### 2.11. Cloud Computing

Cloud computing refers to cost-effective hosting and delivery of computing services, inclusive of servers, databases, storage, networking, software, analytics, and intelligence, using the internet (‘the cloud’) to deliver these services on demand and as required by the organisation [65]. In the context of managing disasters, this would mean that access to data required to manage events would be stored separately from the infrastructure impacted by the disasters and, therefore, remain accessible for collaborative planning and decision making, provided there is an internet connection. Other concerns that need to be addressed include bandwidth requirements to access cloud services and cybersecurity [66]. The advantage is that the data are protected when, e.g., floods or fires threaten the central infrastructure [66]. Another example where cloud services have demonstrated value is in combining wireless sensory networks in real environments to create three-dimensional virtual environment models. These models then provide planners and decision makers with information on the incident in the real environment [66]. In the setting of the COVID-19 pandemic, the Health 4.0 paradigm has been put forward as a way to manage the complexity and assist in diagnosis. Essential components of this paradigm include the concept of ‘Network Interconnection’, which, in turn, has both 5G and cloud computing as essential elements [67]. Recently, a framework was proposed combining the IoT, cloud computing and big data to facilitate access to reliable data from diverse sources to assist analytical calculations in the decision-making process required for preparation, disaster management, and subsequent recovery [68].

### 2.12. Social Media and Internet

Patients and consumers increasingly use the internet and social media to obtain health-related information, including healthcare advice [69]. Researchers have used social media in the context of real-time syndromic surveillance systems that complement traditional public health surveillance methods to identify potential health threats requiring public health intervention [70]. However, the challenge for public health services in using social media and the internet to benefit public health outcomes presents itself in the form of misinformation and disinformation in the context of natural disasters and increasing mistrust in evidence-based science [71].

Twitter and Facebook have been used to deliver real-time information to the community during disasters, and in some cases were the only sources of information for people with no access to traditional media outlets due to power outages or water damage [72]. Social media also provides the opportunity for communities to share information and develop their knowledge communities. Good Karma Networks [73] are one example of Facebook groups where community members can seek support and share knowledge with one another independent of institutions.

**Table 2 ijerph-20-04542-t002:** Strengths and vulnerabilities of DHTs in the context of natural disasters.

Technology	Uptake	Advantages	Vulnerabilities	References
Mobile phones and devices	Widespread use	Rapid, wide-area communications; efficient individual tracking and identification.	Expensive equipment required. Battery storage limits may impair use over time. Partially dependent on access to electricity. Vulnerable to scams/privacy invasion.	[74] [75] [76]
Electronic health records	Increasing	Highly portable. Provides critical clinical history for displaced/non-communicative patients.	Privacy concerns. Dependent on network availability and electrical power. Identification/documentation information may be unavailable to displaced persons. Requires advance implementation for utility in disaster situations.	[77] [78]
Telehealth; Electronic prescribing; Electronic consultation	Widespread use	Proven effectiveness in COVID-19 pandemic. Usable across multiple platforms (mobile devices, landlines, teleconferencing, internet). Greatly expands healthcare workforce effectiveness, especially in understaffed situations. Highly flexible. Demonstrated clinical utility and good evidence base. Use of radio frequency identification (RFID), barcodes, quick response (QR) codes. Limits errors and improves security and traceability.	Dependent on intact communications infrastructure. Privacy concerns. Subtle clinical details may be obscured. Patient unfamiliarity/digital literacy may compromise effectiveness.	[79] [80] [81]
Artificial Intelligence	Limited at present	Facilitates planning and logistics. Likely to aid diagnosis and care delivery in underserved areas in future. Greatest potential may be in training/simulation/situation analysis.	Susceptible to input bias (i.e., data used for training algorithms may not be applicable to all populations). Concerns about “black box” decision making in clinical situations. Requires extensive advance planning/training/infrastructure for use.	[82] [83] [32]
Robotics	Limited at present	Stand-off operation allows for access to dangerous/confined/inhospitable areas. Facilitates search and rescue. Dedicated clinical/surgical systems can deliver remote care. Can be combined with sensor networks and other technologies. Telemonitoring can augment reach of human carers in understaffed/underserved areas.	Expensive equipment. Infrastructure dependent. Requires highly skilled operators and secure communications. Clinical/surgical robots confined to limited procedures. May not be relevant for post-disaster care.	[54] [55]
Wireless Sensor Networks	Limited at present	Facilitate terrain/environmental monitoring. Can provide critical information for public health decision making. Synergistic with other DHTs (e.g., AI; robotics). Potential for supply chain/logistical monitoring in affected areas.	Limited direct clinical utility for healthcare delivery. May require centralized monitoring networks.	[84]
Drones/Uncrewed vehicles	Increasing	Strong potential for use in monitoring, search and rescue, and supply/logistics in affected areas.	May not be a core DHT component. High skill/training requirement at present. Limited power/battery life.	[57] [56] [58]

## 3. Disaster- and Climate-Resilient Healthcare Systems in the Context of Digital Health

Several gatherings of experts have considered the issue of how to make existing and future healthcare facilities and services resilient to disaster- and climate-change-related impacts. The *Riyadh Declaration on Digital Health* provides digital health recommendations to address the challenges of current and future pandemics [85]. Specifically, the declaration calls for action to create the infrastructure needed for rapid preparedness and global responses to share evidence-based practices, implement data-driven and evidence-based protocols for effective communication, and develop a standard global minimum data set and governance structure for public health data and surveillance systems. Many of these recommendations are transferable to making climate-resilient digital health interventions.

The *WHO’s Sendai framework for disaster risk reduction 2015–2030* and technical guidance guide how to operationalize, simplify, and standardize the collection and reporting of data and the key issues to consider in the collection of health data, the types of data collated, and potential stakeholders to engage [86]. The WHO’s global strategy on digital health 2020–2025 [87] and Fast Health Interoperable Resources (HL7)—a freely available resource [88] that describes international standards for the transfer and exchange of clinical and administrative data between software applications to facilitate seamless communication between healthcare providers—are additional sources of guidance that are useful when considering the development of climate-resilient healthcare systems.

### Disaster Recovery Frameworks

Increasingly, jurisdictions and organisations are developing and implementing a disaster response framework that identifies different phases of a disaster and the roles, responsibilities, and processes at each phase. Any disaster response is comprised of several phases, such as prevention, preparedness, response, and recovery [89]. Due to the increasing prevalence of disasters, there has been a shift to viewing disaster response as less linear, and more of a circular and iterative process. In this model, the response and recovery phases aim to increase community resilience and resources in preparation for the next disaster—this has been termed *resilient recovery* [90].

The 2020 Australian Royal Commission into National Disaster Arrangements has proposed a recovery cycle which includes several phases, including ongoing preparedness and recovery planning, relief and short-term recovery, long-term recovery, and transition, as demonstrated in Figure 1 [91]. The Commission also outlined the need for recovery to be led by, and deeply engaged with, local communities.

While, in combination, these frameworks provide insights and guidance on preparing climate-resilient digital health systems, they require transdisciplinary efforts with strategies going beyond hazard vulnerability assessment. Specifically, these frameworks need to be translated to different sectors of society, such as healthcare, and to have consideration given to how disaster response sits alongside an increasingly digital society and healthcare system. Perhaps the greatest challenge for implementing these frameworks is addressing the needs of disproportionately impacted populations, including historically marginalised and underserved communities, encompassing different social, economic, and public health impacts [92]. The following section examines these challenges.

## 4. Natural Disasters

The WHO defines a disaster as ‘*any occurrence that causes damage, ecological disruption, loss of human life or deterioration of health and health services on a scale sufficient to warrant an extraordinary response from outside the affected community or area*.’ [93]. Increasing global temperatures are perhaps the best-known effect of anthropogenic climate change. The type, frequency, and severity of climate-related disasters are influenced by numerous associated factors, including changes in atmospheric moisture content, seasonal snowpack, local vegetation, and soil salinity; and multiple other factors, including changes in atmospheric moisture content [94], seasonal snowpack [95], local vegetation, and soil conditions [96]. These elements can combine to aggravate predictable, seasonally variant events, such as cyclones, monsoon rains, flooding of river basins, and naturally occurring wildfires. In addition, alterations in atmospheric and terrestrial environmental conditions can combine to produce unpredictable, aberrant disaster events, which may cause widespread destruction [97].

The prolongation and intensification of the natural wildfire season in the western United States and southern Australia could be viewed as intensification of predictable disaster types. In contrast, the emergence of powerful heat waves in previously temperate zones, such as the UK [98], and the contemporaneous wildfires in France, Spain, and Portugal in July and August of 2022, could be considered examples of aperiodic or unpredictable climate-related disasters. Both types of situations are expected to become more prevalent as global average temperatures increase. Consequently, healthcare systems worldwide must carefully consider the regional and local risks of natural disasters associated with climate alteration and prepare appropriate adaptation and mitigation strategies as they develop and implement disaster response frameworks. DHTs will play a significant role in advance preparation, disaster reaction, and management. This is reflected in the use of DHTs on an *ad-hoc* basis by victims of the types of disasters described above and in Table 1 and Table 3, as well as the adoption of DHT infrastructure at the regional and national level, which can facilitate disaster response, e.g., migration to EHRs and widespread use of telehealth technologies. The Medicine Sans Frontiers Reconstructive Surgery Program (RSP) is a great example of a DHT-enabled intervention implemented in a humanitarian setting that may have relevance for managing healthcare during natural disasters. It used a multidisciplinary team, including specialists with expertise in rehabilitation medicine, surgery, prosthetics and orthotics, physical and occupational therapy, and biomedical engineering, to collaborate on the development of personalised prosthetic and orthotic devices using 3D technologies and telemedicine spanning paediatric and adult patients from Iraq, Syria, and Yemen [99].

The benefits of prior investment in DHTs were evident in New York City in the aftermath of post-tropical Hurricane Sandy in October-November 2012. In a study of EHR performance following the storm, Morchel et al. cited reports that New York City hospitals and clinics using EHRs experienced only one instance of lost medical records, whereas losses of paper records were described as ‘widespread’ [100].

## 5. Lessons from the Impacts of Natural Disasters on Digital Health Technology to Deliver Quality Healthcare

### 5.1. Study Methods and Analysis

To identify concrete examples of DHTs supporting the delivery of quality healthcare interventions during natural disasters and those technologies which may assist in preparation for future events, the authors undertook a mixed-methods literature review [101] whereby, as a team, the authors undertook a rapid qualitative literature review, followed by deeper narrative analyses to prepare case studies.

Electronic databases PubMed/MEDLINE, Scopus, Web of Science, and CINAHL were searched iteratively, with no time limit or geographical limits imposed. Only literature reported in English was included due to lack of resources to translate papers published in other languages. Bibliographies of all included studies were also reviewed. The following search string was employed in all databases: (digital health technolog*) OR (digital health) AND (natural disasters) AND (healthcare). All types of studies were included to capture lived experiences of populations impacted by disasters. In addition, media publications reporting DHT issues impacting healthcare systems were examined. The aim of this search strategy was not to achieve an exhaustive search of the literature, which would not be feasible given the breadth of the topic, but rather to find a breadth of literature that would contribute to better understanding of the research area. Table 3 represents studies that the authors judged to offer valuable information on lessons learned in employing DHTs in preparation, response, and recovery efforts associated with natural disasters.

When screening the literature, the authors aimed to identify relevant characteristics of DHTs that either led to successful delivery of healthcare in areas of need, required improvements to meet the needs of the healthcare service, or failed to deliver the intended service. Narrative analysis and group discussions were used to examine information related to case studies using the study aims to guide the analysis. To decide which information to focus on, the team utilized the UNDRR ISC Sendai Hazard Definition and Classification Review Technical Report, which lists 302 hazards grouped into eight clusters: meteorological and hydrological hazards, environmental hazards, extraterrestrial hazards, geohazards, chemical hazards, biological hazards, technological hazards, and societal hazards [102]. This report is accompanied by supplementary hazard information profiles [103]. For the purposes of this review, the team focused on floods, wildfires (bushfires), earthquakes, and severe storms.

### 5.2. Lessons Learned

Through the analysis of the included studies, we identified several themes that contributed to understanding how DHTs can be relied on in the delivery of quality healthcare interventions during natural disasters and assist in preparation and recovery efforts associated with these events. These themes were not exhaustive but also aligned with the frameworks introduced earlier in this review. Together, these case study themes reinforce knowledge gained from the process of conducting real-world DHT projects and/or applying DHTs to the delivery of healthcare in response to natural disasters, such as the management and mitigation of natural disaster health risks in regional and global settings.

#### 5.2.1. Infrastructure

As global warming continues to impact the environment, we are also learning more about how it impacts existing infrastructure, particularly the electricity grid and telecommunications, short-term and long-term. For example, heatwaves reduce the generation efficiency of power grids, increase power transmission and distribution loss, decrease the lifetime of equipment such as power transformers, increase peak power demand, and sometimes force power plants offline [104].

The loss of this infrastructure has an extreme effect on communities and day-to-day business. For example, Hurricane Sandy on the eastern seaboard of the United States knocked out 25% of mobile phone towers. At the same time, the accompanying loss of electricity forced many phone service providers offline, meaning that people could not receive information [105]. Wireless infrastructure, fixed infrastructure, and data centres are at high risk of damage associated with hurricanes, storms, typhoons, heat, and wildfires. High winds and falling trees can knock down above-ground telecommunications towers, poles, telephone lines, and microwave receivers. Hurricanes Maria and Irma destroyed over 90% of mobile sites in Puerto Rico, St Martin, Dominica, and Antigua and Barbuda [106]. In the Australian 2019–2020 bushfires, the smoke and heat caused magnetic resonance imaging and computer tomography scanners to stop working in one hospital [107].

There is a need for healthcare organisations to work closely at a transdisciplinary level to seek innovative solutions to develop digitally enabled care models responsive to climate-related infrastructure pressures, including those caused by natural disasters. However, addressing only the infrastructure issues will not be enough to ensure a climate-resilient healthcare system. Work is also needed to address the digital divide in at-risk communities to ensure they can engage with new, digitally enabled models of care. Similarly, education is needed to uplift digital literacy in the healthcare workforce to ensure it can implement and adapt new models of care to meet community needs [108]. There is also the need to address workforce capacity, knowledge, and engagement with climate-change-related impacts on health and in helping communities prepare to manage those risks [109].

Digitally enabled models of care should also include non-digital pathways that can be quickly deployed if the infrastructure is damaged. Reflecting on the Disaster Recovery Framework [91] should compel a move away from simply rebuilding what was broken, to exploring new approaches to healthcare that will withstand future disasters. Similarly, clinicians and communities need to be empowered to appropriate technologies and service models for their needs in a disaster context. In the next section, we examine more specific cases that may provide insight into what this may involve.

#### 5.2.2. Vulnerabilities and Risks to Delivering Quality Healthcare in Disaster Settings

Disaster environments are complex and quick-changing, with each disaster bringing unique circumstances for those impacted and those responding. The case studies below provide insights into the challenge of using and adapting digital health technologies during and after a disaster.

##### Experience of Using DHTs in the Aftermath of the 2015 Nepal Earthquake

Nepal is highly vulnerable to natural disasters, due to its diverse land topography, active tectonic plates, and extreme variation in climate types, ranging from tundra to tropical across a short latitudinal distance (north–south) of 140 km [110]. A major earthquake (magnitude 7.6) and a series of aftershocks struck Nepal in April 2015, killing 8856 people and injuring 22,309 [111]. The earthquake caused severe damage to existing infrastructure, including transportation routes. It debilitated 90% of the local healthcare systems, impeding evacuation and emergency medical responses in the aftermath of the disaster [112]. Consequently, several international humanitarian organisations extended their services, in coordination with local organisations, to provide emergency healthcare needs and other disaster relief activities.

Actors and stakeholders forming part of both formal and informal health system responses made a broad utilisation of information and communication technologies and other DHTs to address the health needs of victims during the disaster [112,113]. “Doctors for You” volunteers used the messenger app WhatsApp to identify requirements for different types of medical resources, and to analyse the temporal aspects of the healthcare needs required across various stages of the relief operation [112]. Basu et al. described some of the challenges associated with implementing DHTs and relief activities in the aftermath of the earthquake. Issues such as power outages, stress, and fear for the safety and well-being of the relief providers amidst the aftershocks were identified as significant challenges [112].

Similarly, Crane et al. [113] reported that following Nepal’s 2015 earthquake, access to DHTs and the capability of locals to use the available technology for medical care were major barriers to the implementation of DHTs. This study demonstrated that in resource-limited settings, communication and digital technologies were essential for networking, establishing contacts with family members, sharing of information among agencies, expressing needs and resources in the disaster hit areas, and coping with mental health issues [113]. In fact, community members expressed concerns about the failure of authorities and health services delivering agencies from the formal sector (state’s response), demanding that officials place digital technologies at the disposal of health services for use in future emergencies.

##### Australian 2019–2020 Bushfires and 2022 Floods

Australia is a large continent subject to a range of climates, including tropical, desert, and temperate zones [114]. As a whole, Australia is a dry continent, with more than 80% of the continent receiving an annual rainfall of less than 600 mm [115]. Australia’s changing climate has seen an increase in average temperatures over the past 60 years, with increased frequency of hot weather, fewer cold days, changing rainfall patterns accompanied by extreme rainfall events, prolonged droughts, decreased snow precipitation, and changes in ocean acidity and rising sea levels [116]. Much of this fluctuation is attributable to anthropic activity. Projections by the Commonwealth Scientific and Industrial Research Organisation (CSIRO), Australia’s leading scientific research organisation, suggest that these events will become more extreme as the climate changes [116]. Extreme weather conditions believed to be driven by climate change were responsible for the catastrophic 2019–2020 Eastern Australian Bushfires [117] and 2022 Eastern Australian floods [118]. Both events were characterised by loss of healthcare services and infrastructure. In affected regions, medical practices and pharmacies were burned down in bushfires or flooded [119], while others lost power and internet connectivity [120]. Regional hospitals and aged care facilities had to be evacuated, and community residents were displaced. Indigenous health services were also severely affected, with detrimental impacts on the provision of culturally safe care [119].

During the Australian bushfires, emergency evacuation instructions sent via digital means were considered effective, but clinical information systems were considered inadequate. Australia’s national personally controlled EHR, known as My Health Record, was underutilised during the Australian bushfires. Specifically, 44% of all My Health Records held no information about people seeking help in rural services outside of their residence [121]. On the other hand, where access to My Health Records existed, pharmacies with access to electricity and power and My Health Records information on patient medication were able to dispense medications safely [121]. Lal et al. [107] suggest that, in preparing health systems for a more extreme climate in Australia, there must be a focus on creating resilient technologies, including DHTs, to develop climate-resilient healthcare systems. The authors call for optimising PHRs to facilitate increased uptake by the population and healthcare organisations to ensure up-to-date clinical records accessible anywhere at any time [107].

DHT utilization during the 2022 floods is still under evaluation; however, the Australian Digital Health Agency was able to serve residents in flood-affected areas using My Health Records, telehealth appointments, and e-prescribing in those areas [122]. The perceptions of affected community members with regard to DHT effectiveness remains to be assessed, along with requirements for optimization of healthcare delivery in terms of quality.

##### Hurricane Florence, 2018, North Carolina, USA

Grover and colleagues illustrated the advantages conferred by DHTs on demand for emergency services with their evaluation of telemedicine use by shelter evacuees following the landfall of Hurricane Florence [123]. In that situation, evacuees were able to contact a contracted telemedicine provider to provide a preliminary assessment of the need for transport to emergency rooms or urgent care facilities. This service substantially reduced the load on hospital emergency rooms by deferring transport of up to 35% of potential patients who would have otherwise gone to the hospital. An additional benefit was provided in cases where evacuees were otherwise well but required refill of prescription medications, either due to inaccessibility of medications or exhaustion of available supply following evacuation. In this instance, the advantage of the telemedicine service was increased as a result of state legislation providing for prescription refills without medical consultation in declared emergencies. This case demonstrates both the utility of telemedicine services in the immediate aftermath of disasters, as well as the advantage conferred by prior government policies which embrace DHTs, and implementation of the infrastructure before the occurrence of the disaster [123].

**Table 3 ijerph-20-04542-t003:** Lessons learned from DHT applications to deliver healthcare in major natural disasters across the globe.

Study/Study Site	Natural Disaster	DHT Type	Equipment Required	Disaster Stage	DHT Application	DHT Strengths	Implementation Challenges	Lessons Learned
Vo A.H et al., 2010 [124] USA	Hurricane Ike	Telemedicine	Broadband computer networks, video monitors, cell phones.	Post-disaster response phase.	Provide a continuum of care and consultation services to those in need.	Telephone-based service with a greater outreach.	Disaster-related impacts. The long sustainability of the technology requires a secure, web-accessible file server system and the development of network hubs.	Flexibility of data networks is essential for resuming operation. Mobile phones can be used to facilitate healthcare but require ‘how to’ protocols. EMR notes in simple text are transferable between systems. Advance planning to secure critical data ahead of disasters by developing web-accessible file server systems. Develop fault-tolerant networks.
Nicogossian et al., 2011 [125] Armenia	1988 earthquake	Telemedicine	Space Bridge communication infrastructure, video monitors, video recording.	Post-disaster response phase.	Provision of healthcare services to earthquake victims.	Well-established guidelines and protocols.	Safeguarding patient privacy, effective connectivity through telecommunications and internet.	A pre-existing system and connectivity are essential for rapid DHT implementation to meet the needs of disaster victims. The system must be staffed with trained personnel for effective consultation and services.
Callaway et al., 2012 [126] Haiti	2010 earthquake	Mobile Health (iChart mHealth)	Gas-powered generator for electricity, satellite antenna for wireless network, cellular phones.	Post-disaster response phase.	Patient tracking, triage, postoperative care, protection of unaccompanied minors, patient handovers.	Reduced workload, improved patient care, adequate patient triage, and improved patient tracking.	No significant challenges reported.	iChart functioned with or without internet connectivity. Improved service delivery using scalable mobile technology.
Kim et al., 2013 [127] USA	Hurricanes, storms, typhoons and other disasters hitting the US gulf coast	Telehealth	Not reported.	Post-disaster recovery phase.	Provide telehealth services to a disaster-affected population across various health specialties.	Multiple shared challenges and recommendations were identified to support the scalable sustainability of telehealth programs.	Inadequate funding impacted the engagement and implementation. Regulatory challenges (i.e., reimbursement for the uninsured). Lack of guidance on establishment of telehealth policies and procedures. User’s confidentiality. Inadequately trained workforce.	Lack of IT support impacts DHT implementation. Adequate bandwidth and network architecture are essential to good connection. Strong vendor support and equipment testing essential for effective response. The framework is adaptable to future telehealth programs for high service needs with limited resources.
Nagata et al., 2013 [128] Japan (Fukushima)	2011 earthquake	Cloud-based Electronic Health Record (EHR)	Low-bandwidth computer networks, laptops, and portable internet Wi-Fi devices.	Post-disaster response phase.	Increase coordination and communication to enhance medical response in the aftermath of the earthquake and subsequent nuclear disaster.	Low-bandwidth internet was sufficient for EHR implementation. Low-cost intervention.	Internet services required for implementation, data security, and privacy.	Low-bandwidth, low-cost cloud-hosted EHR could perform functions needed to provide safe and quality care. Hospital EHRs need to be connected to a national EHR to permit access to patient data during disasters. Benefits of EHR data accessibility during disasters are likely to outweigh the risks concerning privacy issues. Guidelines to manage privacy concerns regarding a national EHR system need to be developed.
Qadir et al., 2016 [129] Pakistan	2015 flood and earthquake	Telepsychiatry	Not described.	Post-disaster response phase.	Treatment of post-traumatic stress disorder (PTSD).	Active community engagement in the telepsychiatry module.	Not reported	Outcome not assessed. Telepsychiatry could deliver effective services where regular services are interrupted.
Taylor et al., 2017 [130] USA	Hurricane Katrina and Hurricane Harvey	EHR	Laptop	Pre-disaster preparedness and post-disaster response phase.	Safe continuum of care in the face of disasters.	EHR was already set up following Hurricane Katrina, which facilitated service continuation during Hurricane Harvey.	Not reported	High-quality healthcare services were achievable amidst the disaster. Patient portals available to access blood test results and medication prescriptions remotely.
Stasiak et al., 2018 [131] New Zealand	2011 Canterbury earthquake	Computerized cognitive behavioural therapy (BRAVE-ONLINE)	Computer with internet services, telephone.	Post-disaster recovery phase.	Cognitive behavioural therapy for anxiety and PTSD.	The DHT (BRAVE-ONLINE) was a validated tool.	Participants were required to be competent technology (computer and internet) users.	With uninterrupted telecommunication services and electricity supply, DHTs can be successfully implemented to provide remote behavioural therapy services.
French et al., 2019 [132] USA	Hurricane Florence	Telehealth	Devices to support video assessment (video monitors and low-bandwidth connection), and cellular phones.	Pre-disaster preparedness and post-disaster response phase.	Test the applicability of telehealth support to evacuation shelters and emergency medical services.	Could operate in settings with low internet bandwidth. Real-time patient assessment and treatment.	Cellular towers can be oversubscribed, limiting the connectivity.	Effective implementation and testing of DHTs before the storm were achieved with minimal changes to existing infrastructures. Was not used in response phase.
Pasipanodya et al., 2020 [133] USA	2015 California fire	Telemedicine	Tablet with internet connectivity, home blood pressure machine.	Pre-disaster period and immediate response period following the disaster.	Management of spinal cord injury.	Cheap and effective.	Not described	Uninterrupted quality of care was possible amidst the California wildfire and its aftermath. Changes to reimbursement structure is necessary.
Grover et al., 2020 [123] USA	Hurricane Florence	Telemedicine (RelyMD)	Tablet	Pre-disaster preparedness and post-disaster response phase.	Reduction in unnecessary emergency medical service utilization and emergency department visits.	The DHT in existence before the disaster.	Not described	Telemedicine limitations were not reported.
Sago et al., 2020 [134] Croatia	Earthquake following the COVID pandemic	Telepsychiatry	Telephone, computers with internet connections for Skype consultations, headphones, smartphones.	Post-disaster early response phase during the COVID-19 pandemic.	Psychological counselling and psychotherapy.	Participants’ compliance and active engagement during the telepsychiatry sessions.	Management of social and interpersonal aspects of participants in the group counselling sessions. Use of technology by the service seekers and service providers. Burnout during long sessions of artificial/remote contact.	DHT limitations were not reported.
Paratz et al., 2022 [135] Timor-Leste	Flood and population dislocation	Cardiac telehealth service	Video monitor, handheld echocardiograms, mobile phones, and landline phones.	Post-disaster response phase during the COVID-19 pandemic.	Management of cardiac care services.	An effective strategy of critical care service delivery amidst the disaster.	Poor internet connection and less coverage. Repeated cycles of disaster during telemedicine services’ reduced efficiency of service delivery. Difficulty in establishing contacts with service seekers. Financial barriers.	Unreliable internet connection and fixed broadband impacted transmission of echocardiographic images. Zoom link used for clinic appointments. Echocardiographic images shared via email and WhatsApp. Recommend cloud-based system for in-time image optimisation

## 6. Future Directions

Undoubtedly, current digital healthcare systems need to be future-proofed to become climate resilient. This future proofing could be achieved through workforce development and infrastructure and service design mindful of changing climate and healthcare needs in the context of increased risk of natural disasters. Another untapped potential is that of citizen scientists, who are, in most cases, also healthcare system users and stakeholders.

### 6.1. Citizen Science

Citizen science is a broad global movement and methodological approach involving active forms of public participation in scientific knowledge production [136,137]. It can also facilitate environmental education and citizenship [138], including disaster risk reduction, climate justice, mitigation, and adaptation [139,140]. Preceding sections have highlighted some examples of public engagement in tackling disaster response through DHTs and social media. Forms of citizen participation at the intersection of human health and natural disasters have been particularly enabled by the internet, mobile and smartphone devices, and crowdsourcing platforms [141,142].

Crowdsourcing is one citizen science approach that has the potential to greatly increase the extent to which natural disaster and environmental health data can be collected by engaging distributed citizens as sensors or volunteer computing experts, usually aided by mobile applications or phone cameras [141]. More advanced crowdsourced data challenges in global health, such as malaria identification and modelling and COVID-19 emergency management undertaken by online communities, have led to real-time solutions that can guide disease-control strategies in pandemic situations [143,144]. These advancements support individuals and community groups in addressing critical questions about disaster health risks and response.

Citizens are usually the first responders in any crisis and come with local knowledge usually unmatched by official agencies [90]. This was seen in the 2022 Lismore Floods in Australia, where a local business owner used social media to compile the only database of people needing to be rescued [145]. Unlike government and industry, communities usually have to appropriate technologies for their disaster response, meaning these rich data cannot easily flow into official channels and may come at a cost to communities. In Australia, the Royal Commission into Natural Disaster Arrangements [91] acknowledged the need to improve the digital infrastructure and data available in the national disaster response. Part of this response could involve uplifting community data and digital resources to ensure these first responders and the rich data they hold on health and community can directly inform the disaster response, including allowing health services to better plan for the community’s needs.

Citizen science natural disaster projects have been reported worldwide, with increasing frequency within the last decade. A 2019 study identified 106 reported projects across earthquake and flooding events, including post-event participation [146]. For instance, the CrowdMonitor application was developed to assign data-gathering tasks to citizens during different emergencies, such as pluvial flooding [147]; and localised weather smartphone apps have been used to crowdsource high-impact weather data [148]. Citizen science approaches were used to understand community response (e.g., evacuation behaviours) to the devastating Kaikōura earthquake and tsunami warnings in Wellington, Aotearoa/New Zealand [149]. Air pollution from the Australian bushfires of 2019–2020 reached hazardous levels across rural and metropolitan areas [150]. Air quality monitoring apps, such as AirRater, which support environmental and public health data, provide participants with guidance on reducing hazardous environmental exposures, such as from bushfire smoke [151]. Improved air quality assessment in schools based in Sydney, Australia, are part of a citizen-centred urban network of air quality sensors, with students actively collecting and analysing air quality data [152]. Rising high temperatures can also increase mortality rates and other morbidities, particularly impacting vulnerable populations and those in lower socio-economic areas [153].

Sensor data collected by citizen scientists have helped, for instance, predict outdoor and indoor heat stress for low-income housing residents in Australia [153]. Citizen science networks of people acting as sensors, observing and recording environmental health information, can include the use of open platforms, such as OpenStreetMap. OpenStreetMap empowers citizens to collaboratively produce a global picture from openly accessible geographic information; e.g., in the impact assessment of the 2015 Nepalese earthquake [154]. The Local Environmental Observer (LEO) Network (https://www.leonetwork.org) is a web platform, with members consisting of Indigenous people with local knowledge, scientists, and the public. Contributing observational data and photos, LEO Network members in Alaska (US) and globally report on-the-ground observations of unusual phenomena in the environment.

These project examples emphasise the opportunity to engage communities to enhance a system’s capacity to respond to natural disaster warnings to ensure community safety and well-being. To be effective, however, citizen science must be considered integral to disaster risk and response infrastructure [155]. According to Ottinger [156], this ideally requires creating communities of practice that include citizen scientists and disaster–healthcare responders and actively connecting new digital platforms and information to existing infrastructures. There is a commensurate need to ensure equity and decolonizing practices to empower participating individuals and communities of practice with digital literacy, agency, and capacity [157]. The US Centers for Disease Control and Prevention (CDC) is an example of an agency that has supported the design of an online toolkit for disaster preparedness, guiding communities and individuals on effectively implementing disaster citizen science projects [158]. Together, such capabilities, underpinned by digitally enhanced networks, reinforce that public participation can be a critical backbone of disaster preparedness, response, and recovery for the health and well-being of communities.

### 6.2. Climate-Resilient Digital Healthcare

The WHO’s Operational Framework for building climate-resilient health systems [159] and the guidance for *Climate-Resilient and Environmentally Sustainable Health Care Facilities* [160] identify technology and infrastructure as one of the four fundamental requirements to provide safe and quality care regardless of the size of the facility or type of services it provides. The WHO’s *Safe Hospital Initiative* provides guidance on safety, security, and functionality of health infrastructure in relation to extreme weather events and other hazards, although it does not provide guidance for protection of DHT infrastructure [161]. Regarding technology, the focus in these documents is largely on physical structures (i.e., a building’s beams, walls, floors, foundations) and on structural components (i.e., architectural elements, emergency access and exit routes, equipment) of the healthcare facility as a means to maintain the facility open and functional. Information on developing climate-resilient digital healthcare systems remains scarce [14].

Climate-resilient digital healthcare systems should be able to anticipate, respond, cope, and recover from climate-related events such as heatwaves and floods. However, this is yet to be achieved. For example, during the 2022 heatwave, two of London’s largest hospitals lost EMR function due to the loss of IT infrastructure [162]. Floods in Australia’s Lismore region destroyed IT infrastructure in many primary care practices [163]. Research is necessary on how to make DHTs function under extreme climate events while simultaneously minimising their carbon footprint. Further work is needed on developing sustainable DHTs that are repairable and recyclable, with long shelf-life and minimal environmental impact. Connecting DHTs nationally and ensuring interoperability between services and jurisdictions is critical to ensuring a resilient healthcare system for everyone. Education to encourage people to use these technologies is also needed, especially where technologies are being introduced in already resource-strained contexts. Barriers to digital access to care providers pre-, during, and post-disasters also need to be addressed to ensure equitable access to healthcare for those most vulnerable in disaster contexts. These are all areas where future research can progress the themes identified in this review.

Workforce capacity development is essential to building climate-resilient, environmentally sustainable healthcare facilities. Training, including information and knowledge management, is needed to enable the workforce to respond to climate risks and threats [160]. Delivery of effective, sustainable, timely, and safe healthcare in the face of increasingly frequent and severe climate-related events requires a fundamental shift in perceptions about what technologies are relevant for healthcare, and how these systems should be designed, prepared, and deployed. These approaches must meet the needs of the workforce and community that are using them. Healthcare professionals require disaster response training that prepares them to utilise technology effectively to meet the needs of their patients, but also their own needs in a crisis situation.

A focus on user experience design can ensure that digitally enabled models of care meet the needs of both clinicians and communities, especially those under increasing pressure due to ongoing disaster threats. Planning, training, and forward-thinking, in addition to substantial funding, is necessary to ensure that already overstretched, fatigued healthcare providers and systems can respond to prolonged and repeated threats, especially in underserved population centres, which are most vulnerable to these shocks. The significant change brought about by new, digitally enabled models of care should support rather than hinder the healthcare workforce and communities in building resilience in the face of increasing climate-related threats.

It is foreseeable that populations will be displaced and exposed to multiple health threats resulting from cyclones, floods, fires, heat waves, vector-borne and infectious diseases, and similar climate-driven or climate-aggravated events. Addressing inequity gaps and environmental determinants of health during DHT design is another factor needed to create climate-resilient DHTs [14]. The WHO guidelines on digital health interventions help focus on the inequity gap by emphasising reaching vulnerable populations sustainably [87]. It is crucial to consider such factors as the long supply chains, extended training periods required for healthcare workers and disaster responders, and the inevitable fear and confusion which accompany major disasters. Consequently, healthcare workers, local, regional, and national governments, and leaders at all levels will be best served by advance preparation and understanding of the advantages and risks associated with DHTs. Exploring how other sectors have employed digital technologies, for example, in managing complex supply chains, can provide insights into adapting these to the delivery of quality healthcare services during natural disasters. By registering digital innovations through the WHO’s global technology registry platform “Global Digital Health Atlas”, these learnings and practices can also be shared globally [164].

Though many healthcare services may have their own disaster response framework, there is value in developing national or international climate disaster response frameworks for healthcare systems that consider the above-mentioned factors. The Disaster Recovery Framework outlined earlier [91] provides a starting point to consider the different phases in disaster response, and the potential outcomes of resilience and adaption that should drive initiatives in the healthcare system.

## 7. Conclusions

As state and federal governments and healthcare organisations seek to address the increasing pressures from climate-driven environmental changes on the community’s health and the healthcare systems, DHTs and related interventions are increasingly seen as a solution to create climate-resilient healthcare systems. Indeed, globally, the evidence for the utility of DHTs in managing preparation, response, and recovery in the context of natural disasters is increasing. However, a concerted effort must be made to make DHTs themselves climate resilient, reflecting the needs of local communities and culturally appropriate care, as well as the needs of the healthcare sector to effectively respond to natural disasters. Significant challenges also remain in the foreground, such as the digital divide and digital literacy in the community and in the healthcare workforce. For these issues to be collectively addressed, co-designing with all stakeholders, inclusive of citizen scientists, at the interdisciplinary level is a fundamental building block. Critically, stakeholders’ perceptions of the value of DHTs in the provision of quality healthcare during natural disasters need to be further explored, as these lived experiences are likely to identify key gaps that must be tackled to optimise DHT functions to meet healthcare needs during natural disasters.

## Figures and Tables

**Figure 1 ijerph-20-04542-f001:**
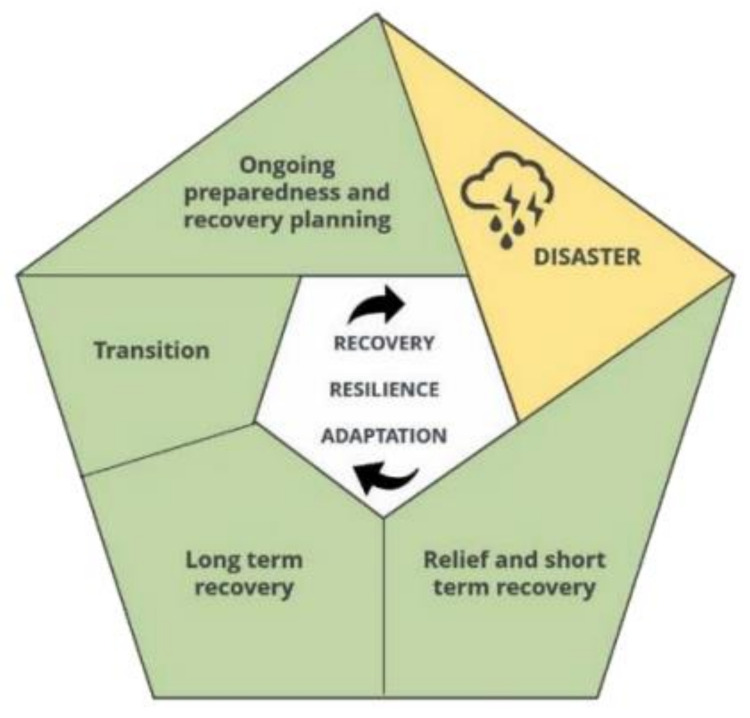
The recovery cycle from the Australian Royal Commission into National Disaster Arrangements 2020 [91], used under a Creative Commons Attribution 4.0 International licence.

## Data Availability

Not applicable.

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
