# Peer review of "Lessons Learned from Natural Disasters around Digital Health Technologies and Delivering Quality Healthcare"

_ijerph, 2023, doi:10.3390/ijerph20054542_

Round 1
Reviewer 1 Report
The study is a scratch attempt to create a protocol for the implementation of digital health technologies in the event of a natural disaster. The authors start from the basis of how the covid situation has used the implementation of these technologies to monitor the pandemic and how the different agents involved have been adapting. It is an interesting study, but of course, it must be taken into account that it is a paper whose objective is to implement a prevention protocol, in a framework that is not legislated and where much of the data may be mere speculation.
Author Response
Thank you for your comment. The aim of our manuscript was to a review of digital health technologies with a focus on their resilience during disaster and climate-change-related impacts. We did not aim to create a protocol or framework for the implementation of digital health technologies in the event of natural disaster. In response to reviewer’s comment we have carefully reviewed the manuscript to ensure that there is no misleading information that would suggest an aim different than the one defined in the introduction of the manuscript, including the removal of Figure 1.
Reviewer 2 Report
This paper presents a review of digital health technologies with a focus on their resilience during disaster and climate-change-related impacts. Within this review, the authors first outline the technologies encompassed by digital health technologies (e.g. telehealth, mobile health, wearables, etc.). Next, they describe disaster recovery frameworks and natural disasters. After this they discuss the impacts on digital health technologies caused by natural disaster, and the lessons that can be learned from them. Finally, they discuss future directions in this space, including citizen science and climate-resilient digital healthcare.
This paper is well written, well organised, and comprehensive. It produces a novel contribution to the field – i.e. that of reviewing digital health technologies with a focus on their resilience during disaster and climate-change-related impacts, and the suggestion of future directions in this space.
My comments are listed below.
(1) Table 3 contains a vast range of natural disaster examples. However, the method of data gathering to obtain this list is not defined, and the studies are not really introduced within the body of the paper. It would be nice to see the method of data gathering included (perhaps at the beginning of Section 5). Section 5 goes into detail about two of the events (2015 Nepal earthquake and 2019-2020 Australian bushfires) but it would be nice to see further events introduced and expanded upon, or the table discussed in more detail as a whole, or both.
(2) Lines 418-420: The same sentence seems to be repeated twice, once with references and once without. I am assuming this is a typo and needs rectifying.
Author Response
Thank you sincerely to the reviewer for the helpful comment. Our response is as follows:
1) We have added information on how the review was performed and data for Table 3 gathered in Section 5. In addition, we have expanded on additional events and the value of Table 2 and 3in more details as a whole, also in section 5.
2) Thank you. The second sentence is now deleted.
Reviewer 3 Report
The article is well-written and its presentation is clear. The topic is relevant and appropriate for this issue and journal. It concisely explains most of the different technologies with applicability in healthcare and their advantages and disadvantages.
I believe it should go forward for publication however I think an explanation could be added about the research methodology used to select the articles discussed and included in the table.
Author Response
Thank you to the reviewer for the helpful comment. In response, we have added information on how the data for Table 3 was gathered in Section 5 - new changes are in red font.
Reviewer 4 Report
1. This paper is interesting and contributes to the development of mission-productive research. For publication in MDPI, I suggest some changes. For detailed comments, please see below:
2. The demonstration of Figure 1. (Digital health technologies in preparation, response and recovery in natural disasters are used by multiple stakeholders to support healthcare systems and healthcare delivery in multiple contexts) is undoubtedly good, but kindly improve the quality of that figure.
3. The formatting of the paper is quite imbalanced, kindly improve it.
4. There are many symbols involved in this article. Please explain the meaning of these symbols one by one.
5. Kindly italicize every single alphabet used in the manuscript (m-Health), it will look more imposing.
6. Kindly cite all references in an MDPI format also, give these references in the same style in the reference section.
7. Lack of proper explanation regarding the result of figures.
8. Check for grammar and punctuation in every single line and paragraph it should be more precise and understandable for the readers.
9. English of the paper should be polished by a professional writer. Some typo errors are found throughout the work, they should be rectified before the final publication of the article.
10. The complete paper is very well written, All the very best (appreciated).
Author Response
Thank you sincerely for very constructive and thoughtful comments. Our response is as follows:
1) Thank you for suggesting these changes, they are valuable.
2) Figure 1 was removed from the text in response to another reviewer's comments
3) In regards to formatting, we have followed journal format as outlined in their template. Should there be any other formatting issues of concern, we will work with the journal managing editor to refine those.
4) All symbols and abbreviations are defined in an article.
5) Regarding suggestions to italicise all uses of terms in the manuscript, because this is not required by the journal formatting we will have reached out to the journal managing editor to seek guidance and will comply accordingly.
6) All references were reported as per journal’s EndNote style. Please note that there will be further work with the managing editors to refine these if needed.
7) Figure 1 was removed from the introductory text as we realise that it may be misleading in terms of its intention. We have also provided a clear in-text introductory sentence to each table and figure to ensure clarity for the purpose of that figure and table.
8) Thank you. The manuscript has been proofread and we have corrected grammar and syntax where required. We will do this again at manuscript proofreading stage.
9) Thank you for this suggestion. We will work with the journal managing editor to see this through.
10) Thank you, we value your feedback which was helpful and thoughtful.
Round 2
Reviewer 4 Report
Manuscript can be accepted for the possible publication.